# Aberrant Splicing Events and Epigenetics in Viral Oncogenomics: Current Therapeutic Strategies

**DOI:** 10.3390/cells10020239

**Published:** 2021-01-26

**Authors:** Flavia Zita Francies, Zodwa Dlamini

**Affiliations:** SAMRC/UP Precision Prevention and Novel Drug Targets for HIV-Associated Cancers (PPNDTHAC) Extramural Unit, Pan African Cancer Research Institute (PACRI), University of Pretoria, Hatfield 0028, South Africa; flavia.francies@up.ac.za

**Keywords:** oncoviruses, aberrant splicing, epigenetic modifications, viral oncogenesis, non-coding RNAs

## Abstract

Global cancer incidence and mortality are on the rise. Although cancer is fundamentally a non-communicable disease, a large number of cancers are known to have a viral aetiology. A high burden of infectious agents (Human immunodeficiency virus (HIV), human papillomavirus (HPV), hepatitis B virus (HBV)) in certain Sub-Saharan African countries drives the rates of certain cancers. About one-third of all cancers in Africa are attributed to infection. Seven viruses have been identified with carcinogenic characteristics, namely the HPV, HBV, Hepatitis C virus (HCV), Epstein–Barr virus (EBV), Human T cell leukaemia virus 1 (HTLV-1), Kaposi’s Sarcoma Herpesvirus (KSHV), and HIV-1. The cellular splicing machinery is compromised upon infection, and the virus generates splicing variants that promote cell proliferation, suppress signalling pathways, inhibition of tumour suppressors, alter gene expression through epigenetic modification, and mechanisms to evade an immune response, promoting carcinogenesis. A number of these splice variants are specific to virally-induced cancers. Elucidating mechanisms underlying how the virus utilises these splice variants to maintain its latent and lytic phase will provide insights into novel targets for drug discovery. This review will focus on the splicing genomics, epigenetic modifications induced by and current therapeutic strategies against HPV, HBV, HCV, EBV, HTLV-1, KSHV and HIV-1.

## 1. Introduction

Cancer is a global burden and is the second leading cause of mortality [1]. It is largely a non-communicable disease attributable to the accumulation of damaged DNA and deleterious mutations in vital genes caused by exposure to carcinogens. Besides, viruses with oncogenic potential are also known to cause cancer through infections. Approximately, 12–20% of all cancers have a viral aetiology [2,3,4]. Oncovirus infections are potentially modifiable risk factors, and targeting infections can be useful in prevention measures. In 2018, the global cancer cases attributable to infections were estimated to be 2.2 million [5]. The International Agency for Research on Cancer (IARC) has identified seven different cancer-causing viruses namely the Human papillomaviruses (HPV), Hepatitis B virus (HBV), Hepatitis C virus (HCV), Epstein–Barr virus (EBV), Human T cell leukaemia virus 1 (HTLV-1), Kaposi’s Sarcoma Herpesvirus (KSHV) and Human immunodeficiency virus 1 (HIV-1). HIV-1 contributes to cancer development through immunosuppression by permitting the co-infection of other oncogenic viruses. With the exception of KSHV, the IARC classified these viruses as group one human carcinogens and further categorised these based on the viral genome as DNA viruses or RNA viruses [6]. HPV, HBV and HCV are major contributors to cancers associated with viral infections (Table 1), and the number of cases varies based on geographic locations. In 2018, Eastern Asia had the highest number of infection-related cancer, with 37.9 cases per 100,000 person-years, closely followed by Sub-Saharan Africa (SSA) with 33.1 cases per 100,000 person-years [5]. A number of these infection-related cancers can be prevented with effective infection control through available vaccines, awareness and understanding of the risk factors.

Viral-induced cancers add a significant burden in low- and middle-income countries in Asia and Africa, particularly in SSA (Table 2). Recent estimates reflect that cancers attributed to HPV infection in Africa are 12.1% and 15.4% in SSA. Although, the estimate of HPV-induced cancers, such as cervical, anus, vulva, vagina, and head and neck, in females in SSA (25.4%) indicate a significant problem [13]. An estimated 50% of liver cancers in SSA are attributed to HBV and 17% to HCV. In comparison, the data arising from North Africa varies slightly with 12% HBV-related cancers and 26% to HCV. KSHV is the most common HIV-1 related cancer infection. In SSA, the estimated Aged Standardized Rates (ASR) for KSHV infections is 32,000 per 100,000 [13]. The estimate for virally-induced cancer in Africa is attributable to EBV infections which is accountable for 2% of all cancer incidences in the continent. Cancers with HIV-1 and HTLV-1 aetiology are estimated to be below 1% in Africa [13]. These incidence estimates are based on current registries, and the numbers may not be an accurate reflection of virally-induced cancers. Furthermore, an under-reporting of cancers in SSA has been observed [14].

Oncoviruses reprogram host molecular pathways to aid viral replication and survival. These alterations facilitate the development of tumourigenesis. They encode oncoproteins that facilitate viral replication and disrupt significant cellular mechanisms that induce aberrant cell proliferation leading to carcinogenesis [15]. Oncoviruses induce cells to evade cell cycle checkpoints and dysregulate apoptosis leading to the accumulation of DNA mutations and genomic instability. Oncoviruses also contribute to epigenetic alterations by interfering with DNA methylation and histone modification to enhance the development of cancer [7]. DNA methylation and histone modification are crucial epigenetic mechanisms that control gene expression and function in DNA repair and replication, growth, and proliferation. Altering DNA methylation and histone modification can have serious cellular repercussions [16]. This review will focus on the splicing genomics and epigenetic modifications induced by HPV, HBV, HCV, EBV, HTLV-1, KSHV and HIV-1. The review will also address the current therapeutic strategies against virally-induced splicing in cancer.

## 2. Molecular Mechanisms of Cancer

### 2.1. Cancer and the Splicing Machinery

Viral oncogenes are causative agents of different cancers, and this is achieved as the virus employs strategies to maintain persistent infection and prolonged latency. Furthermore, the virus also adopts molecular mechanisms that target tumour suppressors and essential signalling pathways. Viral proteins influence the cellular splicing machinery to produce RNA isoforms with carcinogenic properties, resulting in malignancy [17,18,19,20]. Expression of viral oncogenes is regulated by alternative splicing, and therefore, stringent splicing regulation is essential [17]. As a vital post-transcriptional process, alternative splicing is essential to produce several different mRNA isoforms spliced from identical pre-mRNAs that are translated into proteins to facilitate necessary biological functions [21]. Aberrantly spliced isoforms are significantly elevated in cancer cells as compared to healthy cells. A rigorous balance between isoforms with oncogenic characteristics and tumour suppressor activity is essential for cellular cancer surveillance [22,23].

Alternative splicing is a set of processes that is well-defined and regulated. It is initiated by the pre-mRNA that serves as the primary template for alternative splicing. The mRNA is formed through a process known as transcription. The pre-mRNA is formed in a process that is catalysed by RNA polymerase II, leading to the formation of a mature mRNA. Alternative splicing produces mRNA transcripts by excising introns and splicing together the protein-coding exons. Recognition of 5′ and 3′ splice site, and differentiating exons from introns is conducted by a spliceosome which is crucial for efficient and accurate splicing [21,24]. The spliceosome contains five small nuclear ribonucleoproteins (snRNP) particles (U1, U2, U4, U5, and U6 snRNPs) that assemble at the intronic splice sites. In pre-mRNA, exons are defined by the serine/arginine-rich (SR) proteins that outline exon-intron boundaries [24,25]. In contrast, negative regulation of splicing is controlled by heterogeneous ribonucleoproteins (hnRNPs) that generally block exon-intron definition. Therefore, splicing is regulated by the antagonist functions of SR proteins and hnRNPs [22,24]. In the event of a viral infection, the splicing machinery is hijacked to produce viral proteins to maintain the lifecycle of the virus (Figure 1). During infection, viral splicing is regulated by SR proteins and hnRNPs, with elevated expression of these proteins reports in viral-infected cancers [24]. Additionally, alternative splicing plays a role in virally-induced cancer progression by enhancing the expression of proteins to sustain proliferation, evading tumour suppressors and immune response [22].

### 2.2. Cancer and Epigenetic Regulation

Epigenetic modification is a heritable phenomenon that involves molecular modifications that affect gene expression and typically have phenotypic changes. Epigenetic modifications can give rise to cancer as a consequence of histone modification, DNA methylation, modifications in miRNA, inhibition of tumour suppressors, activation of oncoproteins, and infectious agents such as viruses [27,28,29]. Moreover, critical signalling pathways are disrupted by epigenetic modifications that allow the uncontrolled proliferation of malignant cells, apoptotic resistance and distant metastases. Epigenetic modifications are widely prevalent in cancer and could contribute to variations observed in cancer types. Collectively, gene expression of tumour suppressors is altered through epigenetic modifications resulting in carcinogenesis [28].

Chromatin is a well-defined complex that contains DNA and histone octamer that organise the DNA in a compact form. The complex chromatin structure is regulated by DNA methylation [29]. DNA methylation is a process whereby methyl groups are added to the CpG islands, with cytosine bases, at the 5′ end without interfering with the DNA sequence, and the reaction is catalysed by DNA methyltransferases (DNMTs). These DNMTs are DNMT1, DNMT3A and DNMT3B [27,29]. DNA methylation can suppress gene expression when located on the gene promoter region [28]. In cancer cells, these promoters in various genes have hypermethylated CpG islands that suppress gene expression of tumour suppressors [29]. 

The covalent post-translational modification is a process where histones are modified through methylation, phosphorylation, acetylation, ubiquitylation, and SUMOylation. Histone modification is an important biological process in the regulation of transcriptional activation or inactivation, chromosome packaging and DNA repair [27]. The histone octamer includes two copies of H2A and H2B, H3 and H4, with H3 and H4 that are poorly differentiated in cancer cells. Several proteins are essential in the regulation of histone modification namely histone acetyltransferases (HATs), histone methyltransferases (HMTs), protein arginine methyltransferases (PRMTs), histone deacetylases (HDACs), lysine demethylases (KDMs), kinases and phosphatases. Mutations or genetic alterations in these proteins can lead to extensive epigenetic modifications (Figure 1) [28]. Histone modification combined with DNA methylation is vital for gene expression, and their patterns in tumour cells are different compared to healthy cells [27]. Somatic or viral-induced mutations in epigenetic modifiers cause disruption in epigenetic regulations and promote cancer through genomic instability [27,29].

Another aspect of epigenetic regulation are non-coding RNAs (ncRNA). ncRNAs are abundant in the genome with regulatory functions and are divided based on their size as small non-coding RNAs (sncRNAs) and long non-coding RNAs (lncRNAs) [30]. miRNAs, the most characterised sncRNA, downregulate gene expression by binding to mRNA at the 3′ UTR [30]. Similarly, the lncRNAs have various functions such as chromatin regulators and enhancers, and molecular scaffolds. Deregulated lncRNAs through viral infections can restrain the host immune system and affect vital biological processes. Hence, lncRNAs with aberrant expression are reported in various cancers [30,31].

## 3. Virally-Induced Alternative Splicing and Epigenetic Regulation

### 3.1. Human Papillomaviruses and Aberrant Splicing

HPV infection is responsible for a number of cancers with over 90%–100% of high-risk HPV 16 and 18 infections attributed to cervical cancer burden [32]. Of these, HPV 16 infections are the most prevalent and contribute to 55% of cervical cancers [24]. HPV18 is attributed to an estimated 20% of HPV-related cervical cancers, and in combination, HPV16 and 18 contribute to approximately 70% of cervical cancers [32]. The immune system surveillance clears most viral infections; however, recurred infection leads to viral integration into the host genome leads to malignancy [33]. HPV is a circular, double-stranded genome containing three regions known as (i) the long control region (LCR), (ii) the early region (E6, E7, E1, E2, E4 and E5) and (iii) the late region. The LCR is responsible for regulating transcription and replication, while the late region encodes proteins L1 and L2 that function in viral encapsulation through open reading frames (ORFs) [33,34]. The early genes E1 and E2 support viral DNA replication in the host and E2 is deactivated following integration resulting in the expression of E6 and E7. E6 and E7 are important HPV viral oncoproteins with a pleiotropic activity that interfere with host p53 and retinoblastoma protein (pRb), and drive the initiation and progression of cancer. P53 and pRb are vital tumour suppressor genes that function in cell cycle regulation [32]. The p53 degradation is achieved by the interaction of E6 and the ubiquitin-protein ligase known as E6 associated protein (E6AP). HPV-infected cells evade p53-associated cell death and apoptosis by E6 induced inactivation of p53. Similarly, the E7 binds to pRb and promotes its degradation. The expression of cell cycle regulating genes is promoted by the release of E2F transcription factor upon pRb degradation [32]. E6 and E7 splice variants are commonly correlated to high-risk HPV infected cancers (Figure 2) [35].

The central elements for alternative splicing such as transcriptional enhancers and binding sites in HPV are located in the LCR. Early gene expression is transcribed by the early promoter located on the E6 ORF and expression of genes in the late region are carried out by E7 ORF. Gene expression of E6 and E7 in low-risk and high-risk HPVs employ different strategies for transcription. In high-risk HPV, E6 and E7 are transcribed as single polycistronic pre-mRNA from a single early gene promoter. In low-risk HPV, these genes are transcribed using two promoters [35]. Due to alternative splicing of intron E6 in HPV 16, a number of splice variants are formed namely E6*I, E6*II, E6*III, E6*IV, E6*V, E6*VI, E6^E7, E6^E7*I and E6^E7*II. The splice variants identified in HPV 18 are E6*I, E6*II, E6*III, E6^E7 [35,36,37]. Evidence suggests that E6*I and E6*II are high-risk HPV transcripts that are frequently detected in cervical cancer samples in enhanced levels and are associated with advanced disease [38,39,40]. Different variants have been identified in premalignant lesions and HPV-related cancers. Of these, the E6*I splice variant is preferentially generated by splicing 226 and 409 sites. This splice variant has oncogenic functions and higher levels of E6*I mRNA are associated with cervical cancer (Figure 2). However, through the generation of the E6*I variant, a premature stop codon is created and increasing the distance from E6 ORF and E7 ORF that is better suited for efficient E7 translation. Hence, HPV utilises alternative splicing to generate the E6*I splice variant and E7 translation to repress tumour suppressor activity (Figure 2) [35,36,37].

The HPV16 E2 and E6 proteins disrupt RNA splicing by the following mechanisms: (i) The C-terminal of E2 and E6 preferentially bind to the pre-mRNA introns, (ii) E2 and E6 bind to the SR splicing factors—SRSF4, SRSF5, SRSF6 and SRSF9, and other splicing factors, and (iii) the interaction of SR proteins with E2 and E6 occurs at the same site as RNA binding [19,41]. SR proteins regulate alternative splicing by exon-intron recognition. The binding of SR proteins negatively regulates splicing. Henceforth, the binding of E2 and E6 to cellular SR proteins affects the splice site recognition and disrupts the spliceosome assembly, leading to aberrant splicing in the host. Additionally, the N-terminal of E2 is essential to facilitate RNA splicing inhibition along with the C-terminal RNA binding region [41]. SR protein activation is achieved by the serine-arginine protein kinases (SRPK) 1 phosphorylation. Evidence supports that HPV E4 protein binds to SRPK1 and inhibits the activation of SR protein which is crucial in pre-mRNA processing, thereby regulating alternative splicing [24,42,43]. HPV-mediated expression and phosphorylation of splicing factors leads to aberrant cellular splicing and resulting in the production of oncoproteins [19].

### 3.2. Hepatitis B Virus Infection and Dysregulation in Splicing Events

Over 50% of hepatocellular carcinomas have an HBV aetiology with 15–25% of cases developing cirrhosis or liver damage. HBV induces carcinogenesis directly by integrating viral DNA into the host genome, causing mutagenesis and eventually chromosomal instability [44]. The HPV genome encodes four viral genes that are encoded by the HPV genome, which are C (HBcAg), X (HBx), P (DNA polymerase) and S (HBsAg). HBx regulated cellular gene expression is commonly associated with the pathogenesis of hepatocellular carcinoma by promoting cell cycle progression, repressing tumour suppressors and modifying alternative splicing [45]. HBV has four ORF that encodes seven proteins and four unspliced RNAs that are alternatively spliced into numerous transcripts. The four unspliced RNAs are distinguished by size and function: (i) 3.5-kb bicistronic pregenomic RNA (pgRNA), (ii) 2.4-kb preS1 mRNA, (iii) 2.1-kb preS2/S mRNAs and iv) 0.7-kb mRNA. The pgRNA and preS2/S mRNA are major contributors of the splice variants. The pgRNA can be alternatively spliced to generate sixteen variants, while the remaining four variants are generated from alternative splicing of preS2/S mRNA [46]. Of these, the SP1 pgRNA splice variant is frequently detected in approximately 50–60% of patient samples. The pgRNA and preS2/S mRNA splice variants were detected concurrently in patient samples [46]. 

As with many viruses, the splicing of viral RNAs is initiated when HBV hijacks the splicing machinery. These alternatively spliced variants encode proteins that are central to sustained HBV activity in HBV-positive liver tissue. For instance, the polymerase-surface fusion glycoprotein (P-S FP) functions in viral entry, transcription, DNA replication and maturation [36,46]. Alternative splicing of HBV is influenced by environmental factors and liver damage. In patients with chronic HBV infection, elevated levels of SP1 variants and the HBV splicing-generated protein (HBSP) are detected. In chronic infection, the alternative splicing of HBV is regulated by increasing the levels of SP1 pgRNA transcript, which encodes HBSP. HBSP has shown carcinogenic properties such as sustained proliferation, cell transformation, cell migration [47] and the hijacking of the TNF-α signalling pathway, which is vital for apoptosis and inducing immune responses [48]. By blocking the TNF-α signalling pathway, HBSP is responsible for downregulating CCL2 expression, which functions in monocytes and macrophages recruitment elicited by an immune response. As such, HBV preferentially maintains the synthesis of the SP1 variant to increase sustained expression of HSBP to continue blocking the TNF-α signalling pathway to lower the inflammatory response and to evade the adaptive immune response [48,49]. Additionally, HBx is expressed when HBV DNA is integrated into the host chromosome. HBx interacts with several host proteins to regulate biological responses. For instance, HBx interacts with DDB1 protein to initiate degradation of several cellular proteins and modulates DNA damage response resulting in chromosomal aberrations, deleterious mutations and genomic instability that leads to malignancy [50].

New research utilising genome-wide screening has identified alternative splicing events in genes involved in immune system pathway and interferon signalling in HBV and HCV-induced liver cancer. The study provided evidence of differential alternative splicing of human leukocyte antigens (HLA)-A, HLA-C and inositol hexakisphosphate kinase 2 (IP6K2) that are associated with HBV and HCV liver cancer. These proteins play central roles in immune system regulation, malignant transformation, cancer cell migration, invasion and metastasis. HBV and HCV induced different splicing patterns of HLA-A and HLA-C (Figure 3). HBC modifies alternative splicing of HLA-A by increasing intron five retention, and in contrast, HCV produces aberrant isoforms with lower levels of intron five retention. HBV-related hepatocellular carcinoma has elevated levels of HLA-A variants with intron five retention as compared to HCV-related cancers [51]. By modifying the levels of splice variants, tumour cells can evade an immune response [51]. These results highlight the underlying molecular mechanisms utilised by HBV to promote carcinogenesis.

### 3.3. Hepatitis C Virus and Dysregulation in Splicing

HCV, similar to HBV, is also a major risk factor for developing hepatocellular carcinoma. About 5–20% of infected patients develop cirrhosis [53]. HCV, however, does not integrate viral DNA into the host genome. HCV leads to carcinogenesis by regulating aberrant splicing to promote chronic inflammation and cirrhosis [45]. The HCV has an RNA genome with one ORF producing ten viral proteins, namely E1, E2, NS1, NS2, NS3, NS4A, NS4B, NS5A and NS5B. Of these, carcinogenic properties have been detected in NS3, NS4B and NS5A. For HCV to accumulate in HCV-positive liver cells, the 5′-non-coding region of HCV requires interaction with the host miR-122 [36].

Additional to HCV induced different splicing patterns of HLA-A and HLA-C (Figure 3), viral proteins interact with several cellular splicing factors to regulate gene expression. SR protein, a splicing factor, contains arginine-serine rich (RS) domains that are essential in facilitating pre-mRNA splicing. The RS domain is present in several proteins involved in splicing such as the DDX3, a DEAD-box RNA helicase that functions in various cellular processes and RNA splicing [54,55]. DDX3 has shown contradictory cellular functions in various cancers as an oncogene and tumour suppressor by its interaction in several pathways that are modified in cancers, such as the Wnt/β-catenin pathway, E-cadherin pathway, tumour-suppressive miRNA pathway, and interaction with p53 and p21 [56]. HCV interacts with DDX3 at the RS domain and activates the internal ribosome entry site (IRES) translation with differential effects of cellular mRNA splicing. Modified mRNA expression is associated with carcinogenesis. The HCV-mediated splicing regulation through DDX3 primarily occurs in the presence of overexpressed DDX3 that is specific to HCV infection and absent in HBV infection [55,56].

### 3.4. Epstein–Barr Virus and Abnormal Splicing Events

EBV was the first identified herpesvirus, and it is associated with a number of malignancies including B and T cell lymphomas, Burkitt’s lymphoma, leiomyosarcomas, Hodgkin’s lymphoma and gastric cancer [36,57]. An estimated 90% of the world’s population is infected with EBV through mononucleosis infection and are seropositive. EBV life cycle has a latent and a lytic phase during infection. In primary infection, EBV targets B lymphocytes and transforms them into immortal lymphoblasts. In some instances of malignancy, EBV targets T cells and natural killer (NK) cells [36]. EBV encodes six nuclear antigens (EBNA1, 2, 3A, 3B, 3C and EBNA-LP), three latent membrane proteins (LMP1, 2A, and 2B), and the viral BCL-2 homologue, BHRF1. The LMP1 are causative agents that facilitate malignant transformation. B lymphocytes are transformed by the expression of the major oncogenic protein of EBV, LMP1 which is also responsible for triggering multiple cellular signal transduction cascades, along with other EBV proteins such as EBNA1, EBNA2 and EBNA3C [36,57,58]. The BamHI Z fragment leftward open reading frame 1 (BZFL1) is an immediate-early transcription factor of EBV that acts as a switch to activate the lytic phase. By binding to promoters in the lytic genes, BZFL1 triggers signal cascades to promote DNA replication, B cell transformation and evade the adaptive immune response [59,60,61]. Additionally, BZFL1 promotes carcinogenesis by directly binding to p53 through BC–cullin 5–SOCS (BC-SOCS) box ubiquitin-protein ligase complex interaction resulting in p53 degradation (Figure 4) [62]. Recent evidence suggests the carcinogenic properties of LMP2A by the deregulation of signalling pathways, apoptosis and cell cycle [57]. Varying patterns of latent gene expression and coding and non-coding transcripts are related to different cancer types [58]. 

The EBV genome contains two promoters known as the ED-L1 and ED-L1A. Through alternative promoter usage, two LMP1 isoforms are transcribed, (i) the LMP1 ORF is produced by double splicing the ED-L1 pre-mRNA and (ii) the lytic LMP1 (lyLMP1) isoform is a N-terminal truncated version of the LMP1 and is produced by the single splicing of the ED-L1A pre-mRNA [36]. LMP1 sustains the lifecycle of EBV and transforms cell growth and survival patterns by activating MAPK kinase, extracellular signal-regulated kinase, PI3-K/Akt, and signal transducer and activator of three transcription signalling pathways [57]. lyLMP1 is expressed in the lytic phase and functions to negatively regulate LMP1-mediated carcinogenesis [36]. The induction of full-length LMP1 and the transcript lyLMP1 serve different purposes in the life cycle of EBV. Moreover, the lytic phase is negatively regulated in the presence of enhanced LMP1 levels. By regulating the levels of lytic and latent proteins and transcripts through alternative splicing, EBV maintains its life cycle in the host [64]. 

Virus-induced cellular alternative splicing is an important strategy to counteract an immune response and maintain viral protein production and sustain the viral life cycle. EBV RNA splicing is regulated by the trans-acting factor EBV EB2, also known as SM that is expressed in the lytic phase [65]. SM, a family of RNA binding proteins in the lytic phase, has a double-edged post-transcriptional functional role that activates and suppresses gene expression. SM is essential for the post-transcriptional gene expression of EBV, DNA replication and production of the virion [65]. This is achieved by the binding action of SM to RNA to facilitate export and maintain the stability of EBVs mRNA that lacks an intron. SM favours the expression of STAT1β splice variant compared to STAT1α. This way, SM indirectly increases the expression of genes in the interferon signalling pathway that plays an important role in immune responses [66,67]. STAT1 is an essential mediator in the interferon signalling pathway. SM binds to the same region in STAT1 RNA as the splicing factor ASF/SF2 and competes for preferentially binding (Figure 5). Elevated levels of ASF/SF2 impedes SM binding and thereby inhibits SM-induced alternative splicing [67]. This highlights that EBV may be capable of influencing alternative splicing to alter gene expression and sustain its replication in the host cells. In addition, latent infection of EBV influences the alternative splicing of interferon regulatory factor 5 (IRF5) [68]. IRFs are transcription factors that function in immune responses and inflammatory pathways [69]. Evidence shows the production of a novel splice variant, V12, produced by latent EBV infection [68]. Thus, indicating a cellular role of the virus to regulate vital cellular pathways to facilitate latent infection.

A recent study reported the RNA alternative splicing patterns in EBV-associated gastric cancer using transcription-wide analysis. The analysis identified 1297 transcripts that were significantly altered in EBV-positive gastric cancer tissues. Of these EBV-associated transcripts, aberrant splicing events were observed in 77 tumour suppressors, 62 transcription factors and 36 kinases compared to HBV-negative gastric cancer [70]. Additionally, EBV-associated gastric cancer had modified expressions of 67 proteins involved in splicing [70]. The underlying mechanisms leading to aberrant cellular splicing caused by EBV is yet to be elucidated. However, recent reports suggest that EBNA1 could be responsible for modulating alternative cellular splicing [70,71]. It was shown that ENBA1 (i) controls and decreases the expression of splicing factors—SF1, RBM23, hnRNPA1, FOX-2 [71] and SRSF1 [70], (ii) binds to cellular RNA to block its translation and function [71]. These results suggest that EBNA1 may target splicing factors to modulate the pre-mRNA binding function, altering the expression or blocking the binding of splicing factors to regulate transcription of oncogenes [71]. 

### 3.5. Human T Cell Leukaemia Virus 1 and Influence in Aberrant Splicing

HTLV-1 is the first identified retrovirus in adult T cell leukaemia (ATL). Subsequent discoveries reported HTLV-2, -3 and -4, with only HTLV-1 and -2 correlated with human neoplasia. It is estimated that 2%–7% of HTLV-1 carriers are at risk of developing ATL and HTLV-1 infection is significantly associated with several other co-morbidities [72]. The CD4 lymphocytes are predominantly affected by HTLV-1 infection, and malignant transformation occurs through the transcription of a viral oncogene known as Tax. The highly conserved pX region encodes Tax, Rex (an mRNA splicing regulator), the ORF-1 and –II products, namely p8^I^ and p12^I^, and p13^II^ and p30^II^, respectively [73]. These genes are transcribed through alternative splicing [36]. HTLV-1 utilises reverse transcription to initiate replication and integrate DNA into the host genome. All variants are positively transcribed from a single promoter in the 5′ long terminal region (LTR). In comparison, the HBZ is encoded by the antisense RNA that is generated from the 3′ LTR [36]. The viral proteins, HBZ, enhances cell proliferation of the adult T cell leukaemia and associated with leukemogenesis [74,75]. The promoter is located in the LTR where viral transcription occurs. RNA splicing produces eight transcripts—1-E, 1-2-3, 1-2-A, 1-A, 1-2-B, 1-C, 1-3 and HBZ, plus the unspliced transcript [36]. Transcripts with p13^II^ and p30^II^ are essential for integration into the host genome, viral infection and immune evasion [73]. The latter transcript with p30^II^ activates pro-survival signals and enhances proliferation to stimulate lymphoproliferation [73]. Furthermore, since Tax elicits a strong immune response, HTLV-1 stringently regulates Tax expression. Additionally, inflammatory signalling pathways such as NF-κB are stimulated by the HBZ mRNA that negatively regulates Tax-dependent pro-viral gene expression [73,76]. Thus, collectively the mRNA transcripts and proteins produced by the genes in the pX region facilitate the mitotic pro-viral replication of HTLV-1 resulting in lymphoproliferation and evade inflammatory mediators and immune response.

Splicing factors play a central role in alternative splicing and are generally upregulated or downregulated in cancers. SR proteins and hnRNPs are essential modulators of alternative cellular splicing. hnRNPA1, an RNA binding protein, is essential for splice site recognition and facilitates maturation of pre-mRNA transcripts to mRNA [77]. HTLV-1 infected cells have lower levels of hnRNPA1, and evidence has elucidated the interaction between Rex protein and hnRNPA1 [78]. Rex-mediated inhibition of hnRNPA1 could modify the mRNA maturation process that may have serious implications in cellular splicing patterns, increased levels of mRNA with premature stop codons resulting in aberrant protein translation and uncontrolled cell proliferation (Figure 6) [78]. Similarly, ASF/SF2, part of the SR protein family, is essential for splice site selection. It is suggested that the Rex protein may repress the cellular splicing machinery by binding to ASF/SF2 and modifying its function [78]. It is evident that stringent regulation of splice factors is imperative to evade aberrant splicing that could lead to transformation.

Carcinogenesis also occurs through mutated cancer driver genes. Cancer driver genes harbouring deleterious mutations allow the proliferation of tumour cells [79]. RNA sequencing data show that HTLV-1 disrupts cancer driver genes by antisense dependent cis-perturbation or disrupt the premature transcription to drive malignant transformation [80]. HTLV-1 preferentially integrates into the host cancer driver genes at the transcriptionally active regions to prematurely terminate transcription of host genes such as FOXR2, RRAGB, ELF2 and SPSB1. Aberrant transcription of these cancer driver genes leads to oncogenic transformation as a result of immune evasion, survival, proliferation, apoptotic resistance, angiogenesis induction, inhibition of tumour suppressors, and invasion, migration [79,80]. 

### 3.6. Kaposi’s Sarcoma Herpesvirus Infection and Aberrant Splicing Events

KSHV, also referred to as the human herpesvirus 8, is linked to the development of immunosuppressed malignancies such as Kaposi’s sarcoma and B cell lymphoproliferative diseases known as pleural effusion lymphoma and multicentric Castleman disease. KSHV has a linear DNA genome with the virion with highly conserved genes in OFR4-75 and infection persists in a latent and lytic phase. KSHV utilises three promoters LT_c_, LT_d_ and the inducible latent promoter (LT_i_) that acts like a lytic switch [36]. These promoters are responsible for encoding latent viral proteins that have carcinogenic properties; the latent proteins are latent-associated nuclear antigen (LANA), vCyclin (ORF72), vFLIP (K13), Kaposin (K12), interferon regulatory factor 2 (vIRF2–K11.5), vIRF3 (K10.5), and LAMP (K15) [7]. Additionally, KSHV encodes lytic proteins, namely G-protein coupled receptor (vGPCR), vIRF-1 and K1 that are primarily responsible for inducing viral infections [36,81]. 

Gene expression patterns determine the latent and lytic reactivation modes of KSHV infection. Viral latent genes are transcribed from the LT_c_ and LT_d_ promoters. Of the latent genes, LANA is predominantly detected in KSHV infected cells and tumours [81]. KSHV utilises alternative splicing to produce primary RNA transcripts A-G. Typically, through alternative splicing, the unspliced transcript A or E is used to produce LANA and RNA D is the only transcript that is double spliced. Although, in KSHV infected cells, LANA is produced from the A RNA transcript [36]. Malignant transformation of KSHV-infected cells is effectuated by multiple oncogenic functions of KSHV genes and from these alternatively spliced transcripts. For instance, the anti-tumour suppressing activity of the p53 gene and the pRB is inhibited by LANA (LANA1 and LANA2), vCyclin and vIRF-1 (Figure 7) [82]. Furthermore, KSHV inhibits apoptosis by K1 and vIRF-1 induced activation of the NF-kB signal pathway [36], and the vIRF3 gene enhances proliferation and survival of KHSV infected cells is by inhibition of the interferon signalling pathway [81].

Transcriptome analysis of KHSV infected cells reported alternative splicing patterns in 773 genes during early infection. The analysis revealed differential splicing of six insulin receptor genes (IL1B, INSR, IRS2, FOXC2, PRKCA and SOCS7) [84]. Insulin receptor genes exhibit mitogenic effects that function in cell cycle progression, cell proliferation, and have been associated with cancer progression [85]. The mechanisms underlying KHSV-induced cellular alternative splicing are yet to be elucidated. However, KHSV protein interaction and binding with several cellular proteins and splicing factors could provide cues. For instance, KSHV protein ORF57 interacts with the splicing factor, SRSF3. The overexpression of SRSF3 is observed in different cancers [86,87], and the interaction is facilitated by the binding of ORF57 to the N-terminal RNA-recognition motif of SRSF3. By binding to SRSF3, ORF57 inhibits the function of SRSF3 in RNA splicing regulation. Moreover, ORF57 is shown to interact with SRSF1 [88] and hnRNP K [89] to mediate cellular pre-mRNA splicing. Recent evidence shows the K8 bZIP protein encoded by the KHSV virus is an RNA binding protein that coordinates with non-coding RNA to function as a transcriptional repressor, thereby influencing splicing and gene expression of cellular genes [90,91,92]. Lack of stringent regulation of non-coding RNA may lead to carcinogenesis and, the interaction of K8 with non-coding RNAs may drive cellular transformation [93]. By the binding action of ORF57 to cellular splicing factors, KSHV regulates alternative cellular splicing to facilitate cancer-related transcripts and proteins translation. 

### 3.7. Human Immunodeficiency Virus 1-Mediated Splicing Dysregulation

Similar to most pathogenic viruses, HIV-1 regulates its gene expression by exploiting the host splicing machinery. However, unlike oncogenic viruses, malignancy through HIV-1 is achieved through immunosuppression that permits oncogenic virus co-infection. HIV-1 is a positive-strand RNA virus that acts as the mRNA. Using the host cells ribosomes, the HIV-1 RNA directly translates into viral proteins. HIV-1 encodes ten genes with different functions to sustain viral replication and infection [94]. Extensive research of HIV-1 alternative splicing has elucidated numerous spliced variants involved in viral protein transcription and regulation of splicing events [94,95,96,97,98,99]. Through extensive alternative splicing of a primary transcript, HIV-1 produces numerous mRNA isoforms to express viral proteins from nine genes: gag, pol, vif, vpr, vpu, env, nef, rev and tat. The spliced isoforms are categorised as full form, intron-containing variant and intron-less HIV1-1 mRNA. HIV-1 utilises these alternatively spliced transcripts to maintain infection by allowing the maturation of viral particles. Stringent regulation of alternative splicing of these genes is required for sustained HIV-1 infection. The HIV-1 pathogenesis is harshly affected when the RNA export pathway is disrupted. Tat, rev and nef are regulatory proteins that transcribed from completely spliced mRNA in the early stage of infection, whereas vif, vpr, vpu and env are transcribed from partially spliced mRNA following early infection. Lastly, the unspliced mRNA transcribes the structural proteins—gal and pol [95].

Aberrant alternative splicing events are enhanced in HIV-infected individuals that lead to the development of malignancy. Deleterious mutations in central splicing factors can modify the natural cellular function and support carcinogenesis. For instance, evidence has revealed that deleterious mutations in *cis*-acting elements or *trans*-acting factors in angiogenesis genes can affect alternative splicing events and modify cellular signalling pathways to preserve the progression and enhance tumour metastasis (Figure 8) [100]. In HIV-1 related cancers with aberrant splicing events, normal cellular functions may be retained by reversing aberrant splicing by targeting isoforms involved in malignant transformation.

## 4. Viral Infection, Epigenetics and Aberrant Splicing

Viral-induced carcinogenesis is achieved through several strategies that are dependent on the type of virus and the cells they infect. However, a shared feature is that all viruses encode oncoproteins to establish malignancy by overriding essential signalling pathways, apoptosis, disrupting cell cycle control and contributing to uncontrolled cell proliferation. Additionally, RNA sequencing studies have elucidated that viruses alter host mRNA splicing to induce carcinogenesis. The aberrant host alternative splicing could be a result of the virus directly influencing the splicing machinery or through the induction of other host biological responses [101].

Viral infections are capable of inducing changes to cellular alternative splicing events. Alternative splicing is generally linked to transcription, chromatin structure and epigenetics (Figure 9). Hence, modifications in alternative splicing will affect epigenetic and vice versa [101]. The virus commonly targets the spliceosome complex and splicing factors such as the snRNPs, hnRNP and SR proteins and significantly altering its expression. For instance, melanoma differentiation-associated protein 5 (MDA5) and retinoic acid-inducible gene 1 (RIG-I) is vital in the identification of HCV RNA in the cytoplasm, and the elongation factor Tu GTP binding domain-containing protein 2 (EFTUD2) is responsible for mediating antiviral effects [102]. Evidence shows that upon HCV infection, the virus downregulates the expression of EFTUD2 and induced aberrant splicing of RIG-I and MDA5, allowing the virus to evade the cellular antiviral response and facilitate viral replication [102]. In contrast, the HBV oncoprotein Hbx, resulting from the alternative splicing of the X gene, upregulates DNMT1 to repress the expression of host tumour suppressors [7,103]. Therefore, HBV maintains preferential expression of HBx through alternative splicing to promote epigenetic modifications. 

Gene expression is regulated by promoters, enhancers and the accessibility of genes to transcription factors. These genes are located in the chromatin complex, and alternative splicing of these genes coupled to transcription are, therefore, subject to DNA methylation and histone modifications [104]. Gene regulation through epigenetics is important to maintain several biological functions and also relevant to several cancers, including virally-induced cancers. Viruses are known to repress epigenetic regulation of host genes to allow evasion of immune response and additionally, maintain viral latency by suppressing viral genes through epigenetic silencing [104]. Latent infection of oncoviruses is an important strategy in the lifecycle of a virus that leads to malignancy. This is achieved by the integration of the viral genome into the host cell. Subsequent to the integration, viral proteins induce proliferation and transformation. For instance, evidence suggests that host cells DNMT cause de novo methylation of HPV DNA to repress latent infection and viral replication that is causative of cervical cancer [7]. Cervical tumours often show viral DNA hypermethylation, particularly in the LCR and L1 regions of the HPV genome. Cellular transcription factors and the viral E2 proteins bind to the HPV LCR in particular regions called the E2 binding sites (E2BS). E2 protein binding is inhibited upon DNA methylation of E2BS, resulting in enhanced expression of oncoproteins E6 and E7 that drive cervical cancer [7,105]. Additionally, viral E6 and E7 proteins interact with cellular epigenetic proteins and induce histone modification to regulate the expression of tumour suppressors [7,105]. Similarly, the HPV L1 is also hypermethylated in cervical cancers. DNA methylation patterns differ in asymptomatic carriers, precursor lesions and advanced cervical cancers. Therefore, DNA methylation patterns can serve as biomarkers to identify clinical progression caused by HPV infection [7].

Viral-infected cells manipulate epigenetic strategies in favour of immune escape and to repress the expression of tumour suppressors. For instance, EBV positive cells evade immune-related responses through DNA methylation. EBV latent proteins have oncogenic properties, and therefore, the host cytotoxic T cell surveillance recognise EBV latent proteins through the immune response. EBV suppresses the expression of latent proteins histone acetylation and thereby increasing DNA methylation to prolongs its lifecycle and infection [7,106]. Moreover, EBV infection downregulates the expression of several tumour suppressor genes with vital functions in DNA repair, apoptosis and cell cycle regulation through CpG islands methylation that allows the uncontrolled proliferation of lymphoblastoid cell lines [106]. In nasopharyngeal carcinoma cells, the EBV lytic oncoproteins, LMP1 activates DNMT1 to enhance promoter methylation of tumour suppressor genes, and in Hodgkin’s lymphoma, LMP1 enhances expression of an epigenetic activator—KDM6B demethylase [7]. DNA methylation levels steadily increase from asymptomatic infection to carcinogenesis, indicating heightened epigenetic regulation in EBV-induced cancers [7]. Similar epigenetic modulation has been observed in other virally-induced cancers such as B cell lymphoproliferative diseases by KSHV infection. In KSHV, several genes that function in epigenetic inactivation are suppressed by the interaction of LANA and DNA methyltransferase DNMT3a, LANA and TGF-β type II receptor, and KSHV miRNA induces oncogene expression [107]. Reversing the actions of viral-induced epigenetic modifications could provide alternative targets to eradicate infection. 

In general, there are multiple mechanisms that oncoviruses employ to overhaul the splicing and epigenetic machinery to induce carcinogenic effects. Oncoviruses can influence splicing and epigenetics through indirect mechanisms such as introducing mutations in and modulating the expression of splicing factors, post-transcriptional modifications, including phosphorylation and polyadenylation, and modulating the localisation of splicing factors [20,108]. In addition, oncoviruses can regulate epigenetic alternations by inducing histone modification. Alternative splicing of RNA can be affected by histone modification that occurs in close proximity to splice site regions [108]. Consequently, the loss of stringent regulation in RNA splicing and epigenetic mechanisms resulting in carcinogenesis which is attributed to aberrant RNA transcripts that are translated to viral and cellular oncoproteins. 

## 5. Novel Therapeutic Strategies against Virally-Induced Cancers

Expression of viral proteins is necessary to facilitate viral activity. Targeting the expression of central viral proteins may serve as principal targets for novel drug development for viral-induced cancers. As such, aberrant splicing patterns are identified in viral-induced cancers. For instance, RNA-seq data generated through The Cancer Genome Atlas study has revealed alterations in RNA splicing in HBV and HCV-associated hepatocellular carcinoma [45]. In HBV infected liver cancer, 3250 splice variants arising from 2051 genes revealed altered expression. Whereas 1380 splice variants arising from 907 genes had modified expressions in HCV infected liver cancer. A number of these altered transcripts were present in both HBV and HCV related liver cancer. However, 761 transcripts with aberrant splicing were specific to HBV-infection, 68 were specific to HCV-infection, and 299 transcripts were associated with virus-free liver cancer. Therefore, highlighting that viral-induced alternative splicing in cancers can serve as potential candidate biomarkers for diagnosis or targets to pathways for novel therapeutic strategies [45]. Furthermore, recent evidence suggests that HBV transcription is enhanced through the degradation of the Smc5/6 complex attained by the interaction of HBx with cellular DDB1-containing E3 ubiquitin ligase. Targeting HBx-DDB1 interaction may be useful as a novel therapeutic strategy [109]. Similarly, mRNA splicing in HIV-1 infection is regulated by tat and rev. Consequently, targeting these viral proteins to halt alternative splicing, which results in viral protein expression, may render the virus inactive and be targeted for novel drug development [95]. 

Virally-induced alternative splicing modulates gene expression that function is vital cellular and immune responses. The SM protein is essential to maintain EBV viral infection by inducing alternative splicing, and preferentially expresses EBV genes. Recent evidence shows that by blocking the activity of SM and the expression of the corresponding EBV genes, EBV infection can be reversed. For instance, the spironolactone drug blocks SM activity and prevents the synthesis of the viral capsid antigen and thereby impeding capsid formation to inhibit virion production. [110]. In addition, lytic replication of EBV persists by the binding of SM to xeroderma pigmentosum group B-complementing protein (XPB), a transcription factor TFIIH and protein component in DNA repair. Recent evidence shows that spironolactone treatment inactivates XPB rendering it inadequate for SM binding and transcription of EBV genes [111]. Uncovering novel viral-induced alternative splicing and gene expression mechanisms may provide valuable therapeutic targets to control viral-induced cancers.

Along with alternative splicing events, there has been increasing evidence of the association between splicing factors and virally-induced cancer progression. Targeting splicing factors to reverse alternative splicing may have potential therapeutic value. For instance, RBFOX2 was recently identified as a central splicing factor in the differential splicing network, and it is commonly featured in numerous splicing events related to cancer. Evidence shows that RBFOX2 expression is indirectly proportional to the expression of another splicing factor, FKBP2_AP. Thereby suggesting that targeting FKBP2_AP to regulate the expression of RBFOX2 to control its alternative splicing events as potential diagnostic and therapeutic targets for HCV-related cancers is of benefit [112]. 

Epigenetic modification induced by viral infections is another important genetic phenomenon that is prevalent in most cancers with a viral aetiology and specific to tissue type. Epigenetic modifications can be reversible through DNA demethylation and therefore, commonly exploited for novel cancer therapeutic strategies. For instance, DNA methylation regulates normal biological functions that include silencing of repetitive elements in the genome, inhibits the viral sequence integration and transcriptional regulation [28,29]. Aberrant DNA methylation causes genomic instability and can contribute to cancer development. Evidence shows that 5-methylcytosine oxidation to 5-hydroxymethylcytosinecan is achieved by a family of proteins knowns as Ten-eleven translocation (TET). TET functions in the removal of the methyl group and reverse aberrant DNA methylation in cancer cells [27,28].

Microenvironment modifications have been reported in virally-induced cancers. For instance, studies have reported the microenvironment of lymphoma is altered by EBV infection for sustained proliferation and escaping the immune response [113,114]. Alternative splicing events and epigenetic modifications allow viruses to mature and maintain infection, thereby causing changes to the tumour microenvironment. Novel therapeutic strategies can be availed as target candidates by elucidating viral mechanisms and targeting alternatively spliced transcripts that modify the tumour microenvironment for persistent cellular transformation. 

In summary, upon oncoviral infection, viral proteins interact with and bind to numerous splicing factors and epigenetic regulators to induce carcinogenic transformation. Therapeutic inhibitors of these splicing factors and epigenetic regulators may avail as novel drug targets [106,115]. For instance, hnRNPs have a universal role in both alternative splicing and epigenetic modification. Additionally, hnRNPs are also involved in virus-host interaction to sustain viral gene expression [116]. Suppressing the activity of hnRNPs, specific to oncoviruses, may inhibit the interplay between oncoviruses and cellular splicing and epigenetic machinery. Thus, the global transcriptome activity induced by oncoviruses leading to carcinogenesis can be suppressed and may be beneficial as a novel therapeutic strategy [115]. To address this issue, Next Generation Sequencing and transcriptome analysis may provide useful insights to decipher cancer-specific biomarkers involved in epigenetic modulation and alternative splicing in viral-induced cancers as therapeutic approaches. 

## 6. Conclusions

In summary, genetic alterations such as aberrant alternative splicing and epigenetic modifications contribute to the development of carcinogenesis across a range of viral infections with oncogenic potential. Extensive research in virally-induced cancers has elucidated splicing and epigenetic alterations that are specific to cancer and elevated in virally-induced tumours. Therefore, making them a prime target for novel therapeutic strategies. Understanding and unravelling the mechanisms employed by viruses to evade immune surveillance, resist apoptotic signalling and maintaining its lifecycle for latent infection will provide insights that may be fundamental to manipulate RNA splicing and epigenetics to control and treat virally-induced cancer. Furthermore, viruses are capable of hijacking host splicing and epigenetic machinery to regulate protein expression that is favourable for survival, persistent infection and cellular transformation. These are typically achieved by preferential splicing of viral and host mRNA to regulate gene expression through DNA methylation and histone modification. Identifying isoforms and transcriptomic modifications induced by viruses, and their related functions may be potential biomarkers for diagnosis, prognosis and treatment. Novel strategies in diagnosis and treatment will be especially beneficial for low- and middle-income countries in Africa and Asia, that present with a high prevalence of virally-induced cancers. 

## Figures and Tables

**Figure 1 cells-10-00239-f001:**
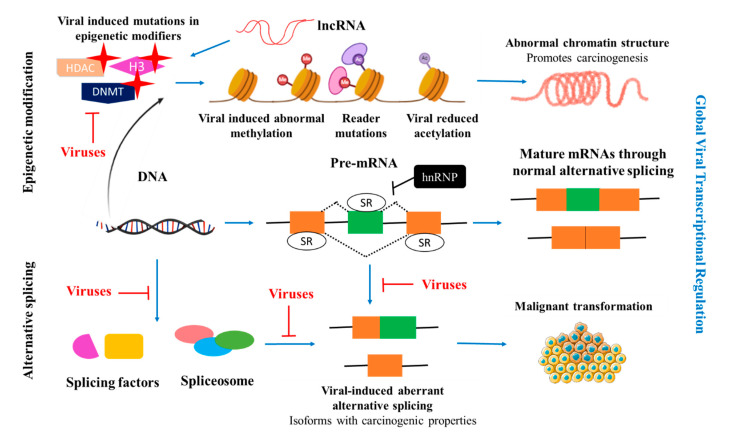
Global virally-induced transcriptional regulation. Viral infections hijack cellular splicing machinery and enhanced epigenetic modifications to promote carcinogenesis through abnormal methylation caused by genetic alterations, histone modification and DNA methylation. Moreover, viral proteins repress the expression and inhibit the binding of splicing factors to modulate aberrant splicing, thereby producing mRNA isoforms with carcinogenic properties that drive cellular transformation [20,26]. hnRNP: heterogeneous ribonucleoproteins; lncRNA: long non-coding RNA; HDCA: Histone deacetylases; DNMT: DNA methyltransferase; H3: Histone H3.

**Figure 2 cells-10-00239-f002:**
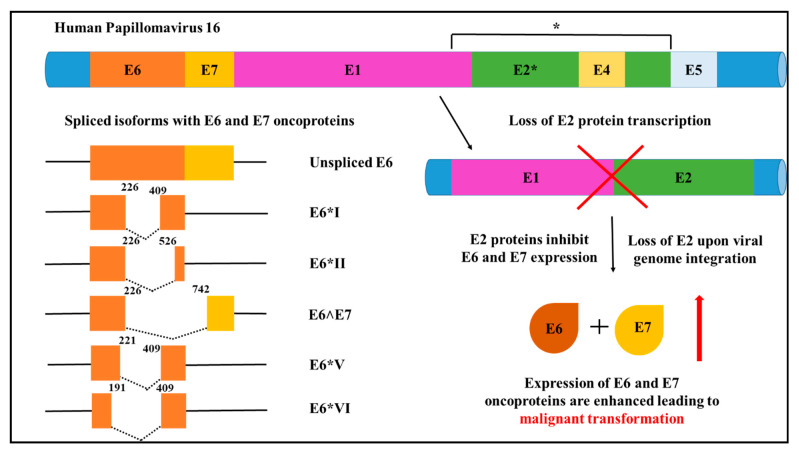
Human Papilloma Virus 16 (HPV16). Major alternative RNA splice variants of HPV16, showing splice variants of E6 and E7 associated with oncogenic properties. Subsequent to viral integration into the host chromosome, E2 transcription is disrupted which enhances oncoproteins E6 and E7 expression to drive malignancy [7,35,36].

**Figure 3 cells-10-00239-f003:**
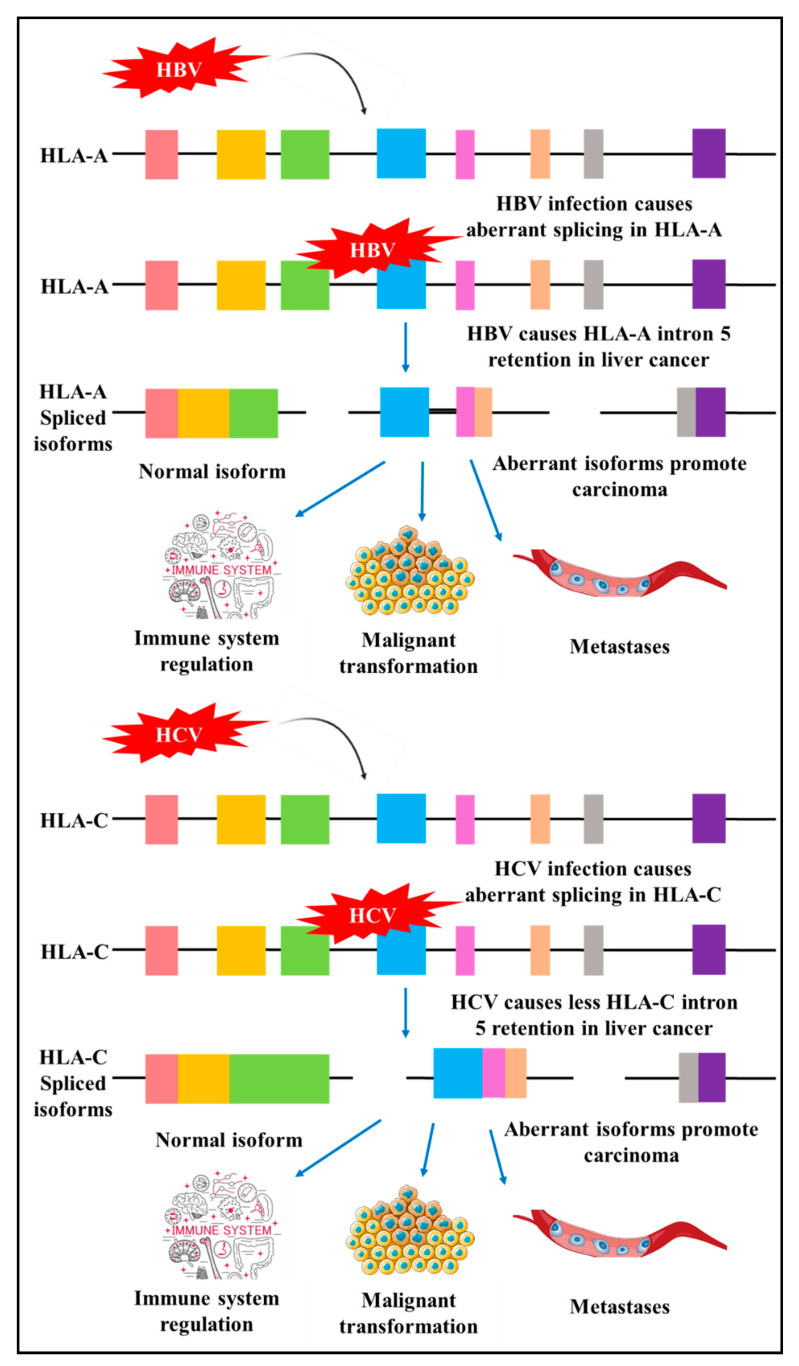
HBV and HCV induced aberrant alternative splicing. HBV disrupts cellular alternative splicing to produce human leukocyte antigen (HLA)-A isoforms with intron five retention compared to HCV. HCV infection leads to infrequent HLA-C intron five retention resulting in malignancy. HLA-A and -C are important in immune surveillance. Coloured boxes: Exons, Black solid line: Introns [51,52].

**Figure 4 cells-10-00239-f004:**
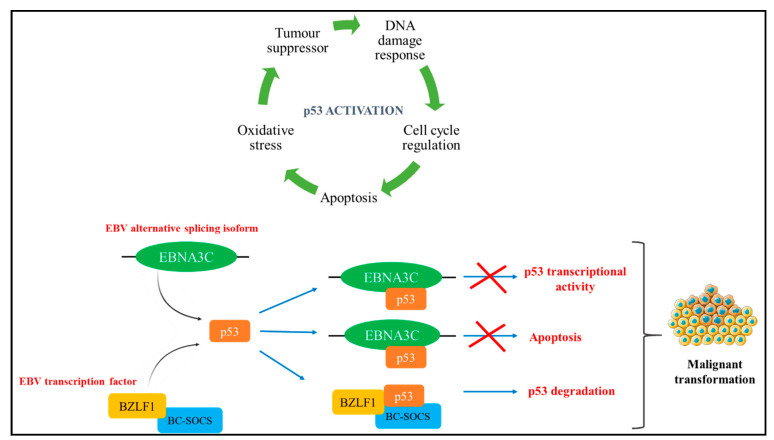
Epstein–Barr virus (EBV) isoform interaction with p53. The EBV spliced isoforms EBNA3C and the transcription factor BZFL1 both interact with p53 to induce oncogenic transformation such as apoptosis, inhibiting the transcriptional activity and degradation of p53 [62,63].

**Figure 5 cells-10-00239-f005:**
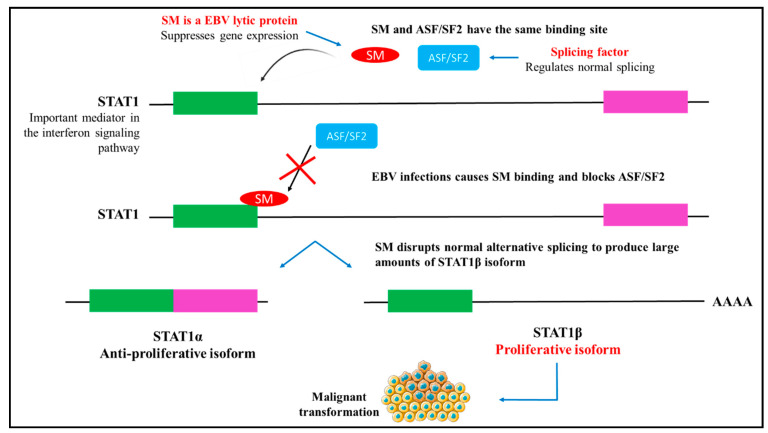
STAT1 and SM (a family of RNA binding proteins in the lytic phase) binding. SM and ASF/SF2 bind to the same region in STAT1. By binding to STAT1, SM regulates alternative splicing and disrupts the ratio of STAT1α and STAT2β isoforms by preferentially increasing the levels of STAT1β. STAT1α and β are principle mediators of the interferon signalling pathway to regulate proliferation, apoptosis and differentiation [67].

**Figure 6 cells-10-00239-f006:**
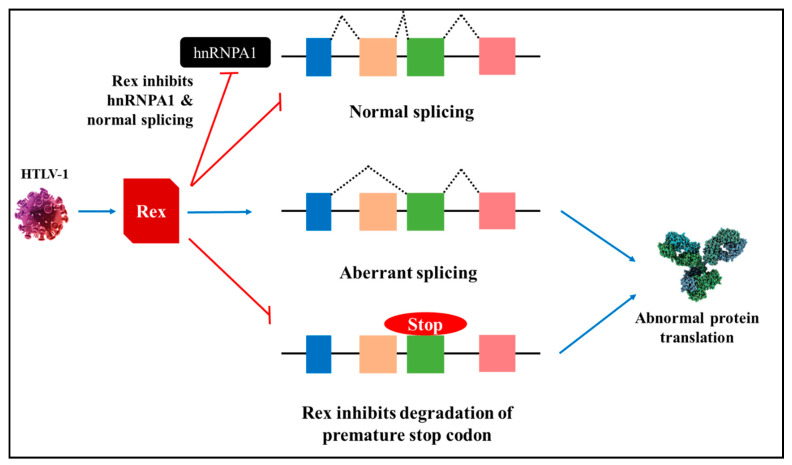
Rex-mediated splicing. Human T cell leukaemia virus 1 (HTLV-1) produced viral protein, Rex, suppresses hnRNPA1 to induce aberrant splicing and inhibit degradation of transcripts with a premature stop codon, thereby leading to abnormal protein translation. Red bars: suppression [78].

**Figure 7 cells-10-00239-f007:**
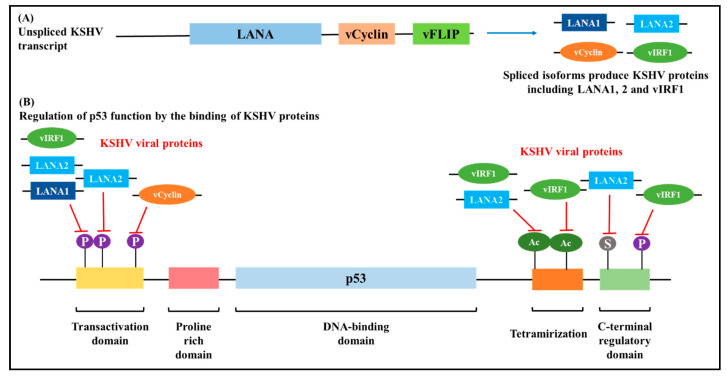
Post-transcriptional regulation of p53 by Kaposi’s Sarcoma Herpesvirus (KSHV) viral proteins. (**A**): Viral RNA isoforms produce several proteins that bind to p53 in different regions and suppress its activity. (**B**): p53 phosphorylation is regulated by the binding of vCyclin, LANA1, LANA2 and vIRF1 at different regions. LANA1, LANA2 and vIRF1 inhibit the phosphorylation whereas vCyclin induces it. p53 protein plays vital roles in cell cycle regulation, and with loss of function of p53, KSHV induces carcinogenesis [36,82,83].

**Figure 8 cells-10-00239-f008:**
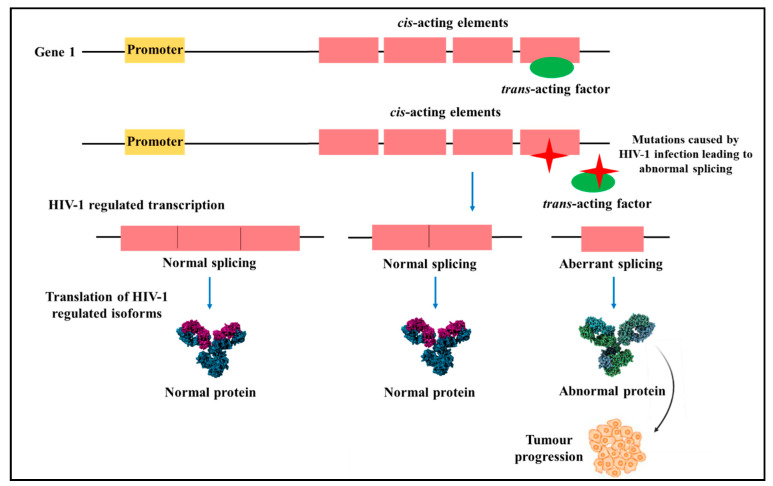
Mutations in *cis*-acting elements and *trans*-acting factors caused by Human immunodeficiency virus 1 (HIV-1). HIV-1 infections lead to deleterious mutations in *cis*-acting elements and *trans*-acting factors in prominent cellular genes leading to abnormal RNA splicing and abnormal protein translation [100].

**Figure 9 cells-10-00239-f009:**
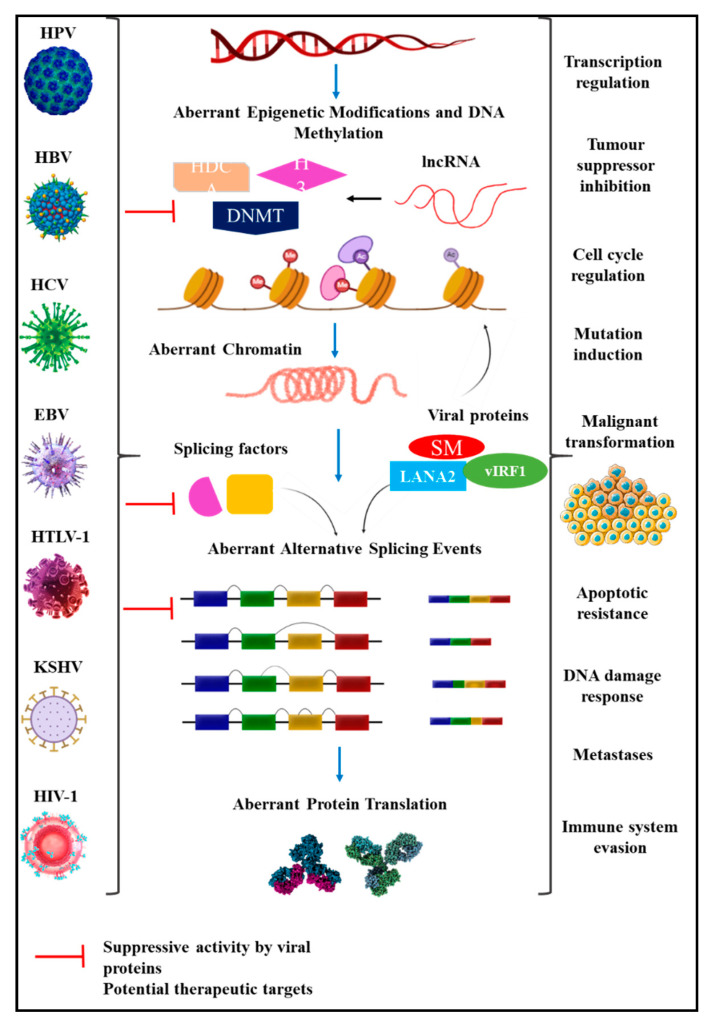
Summary of virally-induced carcinoma through global transcriptomic regulation. HPV, HBV, HCV, EBV, HTLV-1, KSHV and HIV-1 hijack epigenetic and splicing machinery to produce viral proteins and induce cellular changes. These proteins bind to cellular transcriptional factors and tumour suppressors to inhibit biological activities, aiding malignant transformation by modulating transcription resulting in cell cycle regulation, inducing deleterious mutations, resistance to apoptosis and metastases [20,26].

**Table 1 cells-10-00239-t001:** Virally-Induced Cancers and the Associated Viral Aetiology.

Virus	Viral Genetic Material	Cancer Type
Human papillomaviruses	dsDNA	Cervical cancer, anal cancer, vulvar and vaginal cancer, penile cancer, head and neck cancer, skin cancer
Hepatitis B virus	dsDNA	Liver cancer
Hepatitis C virus	ssRNA	Liver cancer
Epstein-Barr virus	dsDNA	B and T cell lymphomas, Burkitt’s lymphoma, leiomyosarcomas, Hodgkin’s lymphoma, post-transplant lymphoproliferative diseases, nasopharyngeal cancer, gastric cancer
Human T cell leukaemia virus 1	dsDNA and ssRNA	Adult T cell leukaemia
Kaposi’s Sarcoma Herpesvirus	dsDNA	Kaposi’s sarcoma, pleural effusion lymphoma, multicentric Castleman disease
Human immunodeficiency virus 1 *	ssRNA	Cervical cancer, Kaposi’s sarcoma, skin cancer, B cell lymphomas

* Carcinogenesis through HIV infection is induced by immunosuppression. dsDNA: double-strand DNA; ssRNA: single-strand RNA [7,8,9,10,11,12].

**Table 2 cells-10-00239-t002:** Virally-Induced Cancers in Africa and Sub-Saharan Africa.

Virus	Africa	Sub-Saharan Africa
Human papillomaviruses	12.1%	15.4%
Hepatitis B and C virus *	2.9%	2.7%
Epstein–Barr virus	2.0%	1.9%
Human T cell leukaemia virus 1	0.1%	0.2%
Kaposi’s Sarcoma Herpesvirus	3.1%	4.2%
Human immunodeficiency virus 1	0.6%	0.8%
Total	20.8%	25.2%

* Collective data for hepatitis B virus (HBV) and hepatitis C virus (HCV) [13].

## Data Availability

Not applicable.

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
