# Peer review of "Aberrant Splicing Events and Epigenetics in Viral Oncogenomics: Current Therapeutic Strategies"

_cells, 2021, doi:10.3390/cells10020239_

Round 1
Reviewer 1 Report
RE: Cells-1050867
This is an excellent review that summarizes virus-induced splicing events in their hosts. The manuscript is well organized and written, with appropriate illustrations. I have only a few minor issues, which can be easily fixed before acceptance.
- A given virus induces unique viral and cellular splicing events. The review lists sufficient splicing events of viral products, but for cellular products, it is a challenge since limited findings have been reported. Some of these findings can be included. For example, EBV latent infection induces different variants of IRF5.
- Please check thorughout the manuscript, some words need to be revised. For example, Line 344. “EBV tissue” should be “EBV-positive gastric cancer tissues”
- Subtitle numbers are incorrect. 1-6. The current numbers are all 1
Author Response
Reviewer 1:
This is an excellent review that summarizes virus-induced splicing events in their hosts. The manuscript is well organized and written, with appropriate illustrations. I have only a few minor issues, which can be easily fixed before acceptance.
- A given virus induces unique viral and cellular splicing events. The review lists sufficient splicing events of viral products, but for cellular products, it is a challenge since limited findings have been reported. Some of these findings can be included. For example, EBV latent infection induces different variants of IRF5.
Response: The reviewer is right in stating that the literature on viral-induced cellular products is limited. However, the following sentences were added in line 335 as follows:
“In addition, latent infection of EBV influences the alternative splicing of interferon regulatory factor 5 (IRF5) [68]. IRFs are transcription factors that function in immune responses and inflammatory pathways [69]. Evidence shows the production of a novel splice variant, V12, produced by latent EBV infection [68]. Thus, indicating a cellular role of the virus to regulate vital cellular pathways to facilitate latent infection.”
- Please check throughout the manuscript, some words need to be revised. For example, Line 344. “EBV tissue” should be “EBV-positive gastric cancer tissues”
Response: The manuscript was checked and the following were amended.
Line 348 – “EBV tissue” was changed to “EBV-positive gastric cancer tissues”
Line 230 – “HBV-positive”
Line 270 – “HCV-positive”
Line 424 – “KSHV-infected cells”
- Subtitle numbers are incorrect. 1-6. The current numbers are all 1
Response: All the subtitle numbers have been amended from 1 to 6.

Reviewer 2 Report
This is a very nice review, with highly relevant references. Please, have a look at this one
Candotti, D.; Allain, J.-P.J.A.o.B. Biological and clinical significance of hepatitis B virus 772RNA splicing: an update. 2017 2017, 2
Author Response
Reviewer 2:
This is a very nice review, with highly relevant references. Please, have a look at this one:
Candotti, D.; Allain, J.-P.J.A.o.B. Biological and clinical significance of hepatitis B virus 772RNA splicing: an update. 2017 2017, 2
Response: We’d like to thank the reviewer for reading this manuscript and providing an input. The suggested reference was already included in the manuscript as reference number 46 and cited in lines 225 and 227.
